# Predictive Information Accelerates Learning in RL

**Kuang-Huei Lee**
Google Research
leekh@google.com

**Ian Fischer**
Google Research
iansf@google.com

**Anthony Z. Liu**
University of Michigan
anthliu@umich.edu

**Yijie Guo**
University of Michigan
guoyijie@umich.edu

**Honglak Lee**
Google Research
honglak@google.com

**John Canny**
Google Research
canny@google.com

**Sergio Guadarrama**
Google Research
sguada@google.com

## Abstract

The *Predictive Information* is the mutual information between the past and the future, $I(X_{\text{past}}; X_{\text{future}})$. We hypothesize that capturing the predictive information is useful in RL, since the ability to model what will happen next is necessary for success on many tasks. To test our hypothesis, we train Soft Actor-Critic (SAC) agents from pixels with an auxiliary task that learns a compressed representation of the predictive information of the RL environment dynamics using a contrastive version of the Conditional Entropy Bottleneck (CEB) objective. We refer to these as Predictive Information SAC (PI-SAC) agents. We show that PI-SAC agents can substantially improve sample efficiency over challenging baselines on tasks from the DM Control suite of continuous control environments. We evaluate PI-SAC agents by comparing against uncompressed PI-SAC agents, other compressed and uncompressed agents, and SAC agents directly trained from pixels. Our implementation is given on GitHub.[1]

## 1 Introduction

Many Reinforcement Learning environments have specific dynamics and clear temporal structure: observations of the past allow us to predict what is likely to happen in the future. However, it is also commonly the case that not all information about the past is relevant for predicting the future. Indeed, there is a common Markov assumption in the modeling of RL tasks: given the full state at time $t$, the past and the future are independent of each other.

However, in general not all RL tasks are specified with a full state vector that can guarantee Markovity. Instead, the environment may be only partially observable, or the state may be represented in very high dimensions, such as an image. In such environments, the task of the agent may be described as finding a representation of the past that is most useful for predicting the future, upon which an optimal policy may more easily be learned.

In this work, we approach the problem of learning continuous control policies from pixel observations. We do this by first explicitly modeling the *Predictive Information*, the mutual information between the past and the future. In so doing, we are looking for a compressed representation of the past that the agent can use to select its next action, since most of the information about the past is irrelevant

for predicting the future, as shown in [4]. This corresponds to learning a small state description that makes the environment more Markovian, rather than using the entire observed past as a state vector. This explicit requirement for a concise representation of the Predictive Information leads us to prefer objective functions that are *compressive*. Philosophically and technically, this is in contrast to other recent approaches that have been described in terms of the Predictive Information, such as Contrastive Predictive Coding (CPC) [30] and Deep InfoMax (DIM) [19], which do not explicitly compress.

Modeling the Predictive Information is, of course, insufficient to solve RL problems. We must also provide a mechanism for learning how to select actions. In purely model-based approaches, such as PlaNet [18], that can be achieved with a planner and a reward estimator that both use the model's state representation. Alternatively, one can use the learned state representation as an input to a model-free RL algorithm. That is the approach we explore in this paper. We train a standard Soft Actor-Critic (SAC) agent with an auxiliary model of the Predictive Information. Together, these pieces give us Predictive Information Soft Actor-Critic (PI-SAC).

The main contributions of this paper are:

- **PI-SAC:** A description of the core PI-SAC agent (Section 3).

- **Sample Efficiency:** We demonstrate strong gains in sample efficiency on nine tasks from the DM Control Suite [42] of continuous control tasks, compared to state-of-the-art baselines such as Dreamer [17] and DrQ [27] (Section 4.1).

- **Ablations:** Through careful ablations and analysis, we show that the benefit of PI-SAC is due substantially to the use of the Predictive Information and compression (Section 4.2).

- **Generalization:** We show that compressed representations outperform uncompressed representations in generalization to unseen tasks (Section 4.3).

## 2 Preliminaries

**Predictive Information.**   The *Predictive Information* [4] is the mutual information between the past and the future, $I(\text{past}; \text{future})$. From here on, we will denote the past by $X$ and the future by $Y$. [4] shows that the entropy of the past, $H(X)$, is a quantity that grows much faster than the Predictive Information, $I(X; Y)$, as the duration of past observations increases. Consequently, if we would like to represent only the information in $X$ that is relevant for predicting $Y$, we should prefer a *compressed* representation of $X$. This is a sharp distinction with techniques such as Contrastive Predictive Coding (CPC) [30] and Deep InfoMax (DIM) [19] which explicitly attempt to maximize a lower bound on $I(X; Y)$ without respect to whether the learned representation has compressed away irrelevant information about $X$.

**The Conditional Entropy Bottleneck.**   In order to learn a compressed representation of the Predictive Information, we will use the Conditional Entropy Bottleneck (CEB) [7] objective. CEB attempts to learn a representation $Z$ of some observed variable $X$ such that $Z$ is as useful as possible for predicting a target variable $Y$, i.e. maximizing $I(Y; Z)$, while compressing away any information from $X$ that is not also contained in $Y$, i.e. minimizing $I(X; Z|Y)$:

$$CEB \equiv \min_Z \beta I(X; Z|Y) - I(Y; Z) \tag{1}$$

$$= \min_Z \beta(-H(Z|X) + H(Z|Y)) - I(Y; Z) \tag{2}$$

$$= \min_Z \mathbb{E}_{x,y,z \sim p(x,y)e(z|x)} \beta \log \frac{e(z|x)}{p(z|y)} - I(Y; Z) \tag{3}$$

$$\leq \min_Z \mathbb{E}_{x,y,z \sim p(x,y)e(z|x)} \beta \log \frac{e(z|x)}{b(z|y)} - I(Y; Z) \tag{4}$$

Here, $e(z|x)$ is the true *encoder* distribution where we sample $z$ from; $b(z|y)$ is the variational *backwards encoder* distribution that approximates the unknown true distribution $p(z|y)$. Both can be parameterized by the outputs of neural networks. Compression increases as $\beta$ goes from 0 to 1.

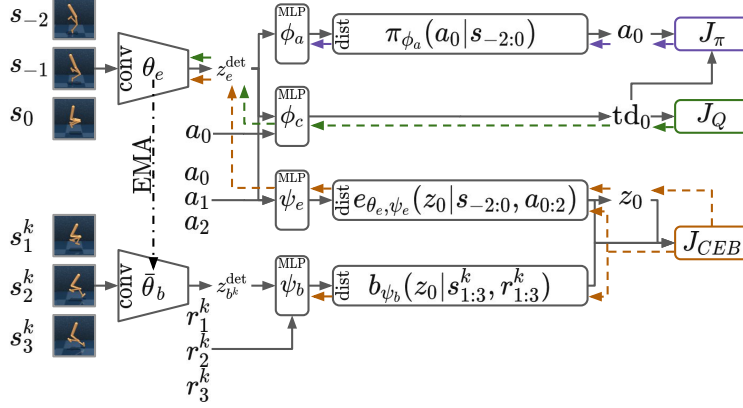

Figure 1: PI-SAC system diagram for a single minibatch example. To compute $J_{CEB}$ requires $K$ $b_{\psi_b}(\cdot)$ distributions from the minibatch, as described in Section 2. Colored edges show how gradients flow back to model weights.

To get a variational lower bound on the $I(Y;Z)$ term, we will use the *CatGen* formulation from [7], which is equivalent to the InfoNCE bound [30, 32] but reuses the backwards encoder:

$$I(Y;Z) \geq \mathbb{E}_{x,y,z \sim p(x,y)e(z|x)} \log \frac{b(z|y)}{\frac{1}{K}\sum_{k=1}^{K} b(z|y^k)} \quad (5)$$

We write the objective for a single example in a minibatch of size $K$ to simplify notation. The $K$ examples are sampled independently. Altogether, this gives us:

$$CEB \leq \min_{Z} \mathbb{E}_{x,y,z \sim p(x,y)e(z|x)} \beta \log \frac{e(z|x)}{b(z|y)} - \log \frac{b(z|y)}{\frac{1}{K}\sum_{k=1}^{K} b(z|y^k)} \quad (6)$$

**Soft Actor-Critic.** Soft Actor-Critic (SAC) [16] is an off-policy algorithm that learns a stochastic policy $\pi_{\phi_a}$, a Q-value function $Q_{\phi_c}$, and a temperature coefficient $\alpha$ to find an optimal control policy. It maximizes a $\gamma$-discounted return objective based on the Maximum Entropy Principle [45, 44, 15, 24]. SAC has objectives for the critic, the actor, and the temperature parameter, $\alpha$. The critic minimizes:

$$J_Q(\phi_c) = \mathbb{E}_{s_t,a_t \sim \mathcal{D}} \frac{1}{2}\Big(Q_{\phi_c}(s_t,a_t) - (r(s_t,a_t) + \gamma \mathbb{E}_{s_{t+1} \sim p} V_{\bar{\phi}_c}(s_{t+1}))\Big)^2 \quad (7)$$

where $V_{\bar{\phi}_c}(s_t) \equiv \mathbb{E}_{a_t \sim \pi(a_t|s_t)} Q_{\bar{\phi}_c}(s_t,a_t) - \alpha \log \pi(a_t|s_t)$ is a value function that uses an exponential moving average of the $\phi_c$ parameters, $\bar{\phi}_c$. The actor minimizes:

$$J_\pi(\phi_a) = \mathbb{E}_{s_t \sim \mathcal{D}} \mathbb{E}_{a_t \sim \pi(a_t|s_t)} \alpha \log \pi(a_t|s_t) - Q_{\phi_c}(s_t,a_t) \quad (8)$$

Given $\mathcal{H}$, a target entropy for the policy distribution, the temperature $\alpha$ is learned by minimizing:

$$J_\alpha(\alpha) = \mathbb{E}_{a_t \sim \pi(a_t|s_t)} -\alpha \log \pi(a_t|s_t) - \alpha\mathcal{H} \quad (9)$$

SAC learns two critics $Q_{\phi_c^1}, Q_{\phi_c^2}$ and maintains two target critics $Q_{\bar{\phi}_c^1}, Q_{\bar{\phi}_c^2}$ for double Q-learning [9], but we omit it in our notation for simplicity and refer readers to the SAC paper [16] for details.

## 3   Predictive Information Soft Actor-Critic (PI-SAC)

A natural way to combine a stochastic latent variable model trained with CEB with a model-free RL algorithm like SAC is to use the latent representation at timestep $t$, $z_t$, as the state variable for the actor, the critic, or both. We will call this *Representation* PI-SAC and define it in Section H. However, any representation given to the actor cannot have a dependency on the next action, and any representation given to the critic can depend on at most the next action, since during training and evaluation, the actor must use the representation to decide what action to take, and the critic needs the representation to decide how good a particular state and action are. The CEB model's strength, on the

---

**Algorithm 1** Training Algorithm for PI-SAC
___

**Require:** $E_{\text{step}}, \theta_e, \phi_c^1, \phi_c^2, \phi_a, \alpha, \psi_e, \psi_b$      ▷ Environment and initial parameters
    $\bar{\theta}_b \leftarrow \theta_e, \bar{\theta}_e \leftarrow \theta_e$    ▷ Copy initial forward conv weights to the backward and target critic conv encoder
    $\mathcal{D} \leftarrow \emptyset$      ▷ Initialize replay buffer
    **for** each initial collection step **do**      ▷ Initial collection with random policy
        $a_t \sim \pi_{\text{random}}(a_t)$      ▷ Sample action from a random policy
        $s_{t+1}, r_{t+1} \sim E_{\text{step}}(a_t)$
        $\mathcal{D} \leftarrow \mathcal{D} \cup (s_{t+1}, a_t, r_{t+1})$
    **end for**
    $s_1 \leftarrow E_{\text{step}}()$      ▷ Get initial environment step
    **for** t=1 **to** M **do**
        $a_t \sim \pi_{\phi_a}(a_t|s_t)$      ▷ Sample action from the policy
        $s_{t+1}, r_{t+1} \sim E_{\text{step}}(a_t)$      ▷ Sample next observation from environment
        $\mathcal{D} \leftarrow \mathcal{D} \cup (s_{t+1}, a_t, r_{t+1})$
        **for** each gradient step **do**
            $\{\phi_c^i, \theta_e\} \leftarrow \{\phi_c^i, \theta_e\} - \lambda_Q \hat{\nabla}_{\{\phi_c^i, \theta_e\}} J_Q(\phi_c^i, \theta_e)$ for $i \in \{1, 2\}$      ▷ gradient step on critics
            $\phi_a \leftarrow \phi_a - \lambda_\pi \hat{\nabla}_{\phi_a} J_\pi(\phi_a)$      ▷ gradient step on actor
            $\alpha \leftarrow \alpha - \lambda_\alpha \hat{\nabla}_\alpha J_\alpha(\alpha)$      ▷ adjust temperature
            $\{\theta_e, \psi_e, \psi_b\} \leftarrow \{\theta_e, \psi_e, \psi_b\} - \lambda_{CEB} \hat{\nabla}_{\{\theta_e, \psi_e, \psi_b\}} J_{CEB}(\theta_e, \psi_e, \psi_b)$      ▷ CEB gradient step
            $\bar{\phi}_c^i \leftarrow \tau\phi_c^i - (1-\tau)\bar{\phi}_c^i$ for $i \in \{1,2\}$      ▷ Update target critic network weights
            $\bar{\theta}_e \leftarrow \tau\theta_e - (1-\tau)\bar{\theta}_e$      ▷ Update target critic conv encoder weights
            $\bar{\theta}_b \leftarrow \tau_b\theta_e - (1-\tau_b)\bar{\theta}_b$      ▷ Update backward conv encoder weights
        **end for**
    **end for**
___

other hand, lies in capturing a representation of the dynamics of the environment multiple steps into the future. We may therefore hypothesize that using CEB as an auxiliary loss can be more effective, since in that setting, the future prediction task can be conditioned on the actions taken at each future frame. Conditioning on multiple future actions in the forward encoder allows it to make more precise predictions about the future states, thereby allowing the forward encoder to more accurately model environment dynamics.[2] Consequently, PI-SAC agents are trained using CEB as an auxiliary task, as shown in Figure 1.

PI-SAC uses the same three objective functions from SAC, described above. The only additional piece to specify is the choice of $X$ and $Y$ for the CEB objective. In our setting, $X$ consists of previous observations and future actions, and $Y$ consists of future observations and future rewards. If we define the present as $t = 0$ and we limit ourselves to observations from $-T + 1$ to $T$, we have:

$$J_{CEB}(\theta_e, \psi_e, \psi_b) = \mathbb{E}_{s_{-T+1:T}, a_{0:T-1}, r_{1:T} \sim \mathcal{D}, z_0 \sim e(z_0|\cdot)} \log \frac{e_{\theta_e, \psi_e}(z_0|s_{-T+1:0}, a_{0:T-1})}{b_{\psi_b}(z_0|s_{1:T}, r_{1:T})}$$
$$+ \log \frac{b_{\psi_b}(z_0|s_{1:T}, r_{1:T})}{\frac{1}{K}\sum_{k=1}^{K} b_{\psi_b}(z_0|s_{1:T}^k, r_{1:T}^k)} \quad (10)$$

The training algorithm for PI-SAC is in Algorithm 1. $E_{\text{step}}$ is the environment step function. $\theta_e$ is the weight vector of the convolutional encoder. $\bar{\theta}_b = \text{EMA}(\theta_e, \tau_b)$ is the weight vector of the convolutional backwards encoder, where $\text{EMA}(\cdot)$ is the exponential moving average function. $\phi_c^1, \phi_c^2$, and $\phi_a$ are the weight vectors for two critic networks and the actor network, respectively. We let the critic gradients back-propagate through the shared conv encoder $e_{\theta_e}$, but stop gradients from actor. The target critics use a shared conv encoder parameterized by $\bar{\theta}_e$ which is an exponential moving average of $e_{\theta_e}$, similar to updating the target critics. $\alpha$ is the SAC temperature parameter. $\psi_e$ and $\psi_b$ are the weight vectors of MLPs to parameterize the CEB forward and backwards encoders. $\tau$ and $\tau_b$ are exponents for EMA calls. $\lambda_Q, \lambda_\pi, \lambda_\alpha$, and $\lambda_{CEB}$ are learning rates for the four different objective functions. See Section A for implementation details.

___

[2]We give details and results for Representation PI-SAC agents in Section H. Representation PI-SAC agents are also very sample efficient on most tasks we consider, but they don't achieve as strong performance on tasks with more complicated environment dynamics, such as Cheetah, Hopper, and Walker.

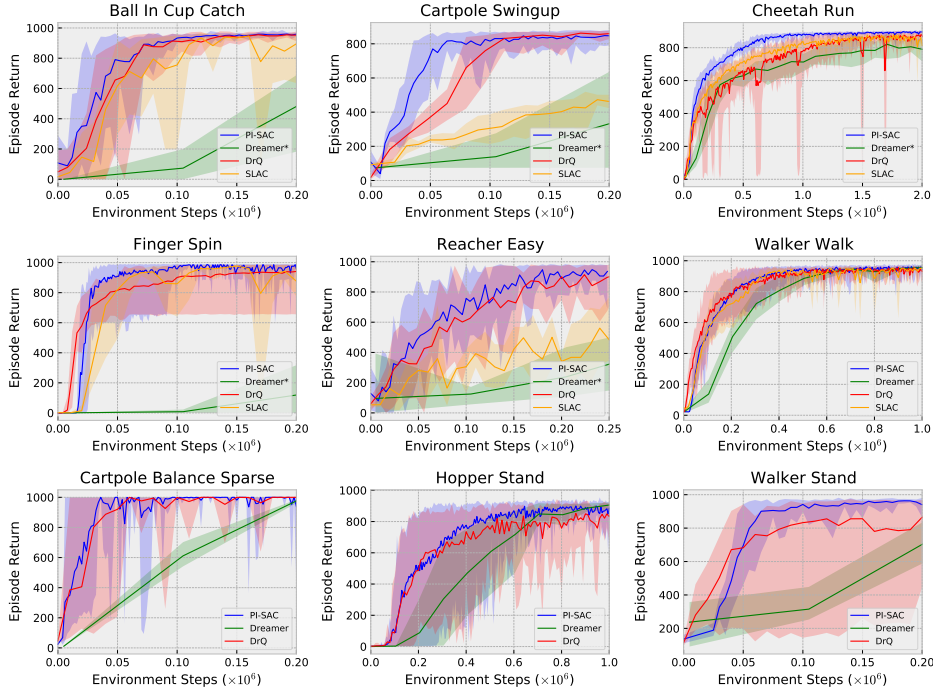

Figure 2: Performance comparison to existing methods on 9 tasks from DeepMind control suite. The upper 6 tasks are the PlaNet benchmark [18]. *Dreamer\** indicates that the other agents do not use Dreamer's action repeat of 2. We additionally include the 3 lower tasks with a fixed action repeat of 2 to compare with Dreamer [17] and DrQ [27] results on the Dreamer benchmark. PI-SAC matches the state-of-the-art performance on all 9 tasks and is consistently the most sample efficient.

# 4 Experiments

We evaluate PI-SAC on the DeepMind control suite [42] and compare with leading model-free and model-based approaches for continuous control from pixels: SLAC [29], Dreamer [17], CURL [39] and DrQ [27]. Our benchmark includes the six tasks from the PlaNet benchmark [18] and three additional tasks: Cartpole Balance Sparse, Hopper Stand, and Walker Stand.

The PlaNet benchmark treats action repeat as a hyperparameter. On each PlaNet task, we evaluate PI-SAC with the action repeat at which SLAC performs the best, and compare with the best DrQ and CURL results. The choices of action repeat are listed in Section A.2. On Walker Walk (also in the PlaNet benchmark), Cartpole Balance Sparse, Hopper Stand, and Walker Stand, we evaluate PI-SAC with action repeat 2 and directly compare with Dreamer and DrQ results on the Dreamer benchmark. We report the performance using true environment steps to be invariant to action repeat. All figures show mean, minimum, and maximum episode returns of 10 runs unless specified otherwise.

Throughout these experiments we mostly use the standard SAC hyperparameters [16], including the sizes of the actor and critic networks, learning rates, and target critic update rate. Unless otherwise specified, we set CEB $\beta = 0.01$. We report our results with the best number of gradient updates per environment step in Section 4.1, and use one gradient update per environment step for the rest of the experiments. Full details of hyperparameters are listed in Section A.2. We use an encoder architecture similar to DrQ [27]; the details are described in Section A.1.

## 4.1 Sample Efficiency

Figure 2 and Table 1 compare PI-SAC with SLAC, Dreamer, DrQ, and CURL[3]. PI-SAC consistently achieves state-of-the-art performance and better sample efficiency across all benchmark tasks, better

| 100k step scores | PI-SAC | CURL | DrQ | SLAC |
|---|---|---|---|---|
| Ball in Cup Catch | **933±16** | 769±43 | 913±53 | 756±314 |
| Cartpole Swingup | **816±72** | 582±146 | 759±92 | 305±66 |
| Cheetah Run | **460±93** | 299±48 | 344±67 | 344±69 |
| Finger Spin | **957±45** | 767±56 | 901±104 | 859±132 |
| Reacher Easy | **758±167** | 538±233 | 601±213 | 305±134 |
| Walker Walk | 514±89 | 403±24 | **612±164** | 541±98 |
| Cartpole Balance Sparse | **1000±0** | - | **999±2** | - |
| Hopper Stand | **97±147** | - | 87±152 | - |
| Walker Stand | **942±21** | - | 832±259 | - |
| 500k step scores | PI-SAC | CURL | DrQ | SLAC |
| Cheetah Run | **801±23** | 518±28 | 660±96 | 715±24 |
| Reacher Easy | **950±45** | 929±44 | 942±71 | 688±135 |
| Walker Walk | **946±18** | 902±43 | 921±45 | 938±15 |
| Hopper Stand | **821±166** | - | 750±140 | - |

Table 1: Returns at 100k and 500k environment steps. We only show results at 500k steps for tasks on which PI-SAC is not close to convergence at 100k steps. We omit Dreamer's results using action repeat of 2 (most of the scores are significantly lower as shown in Figure 2).

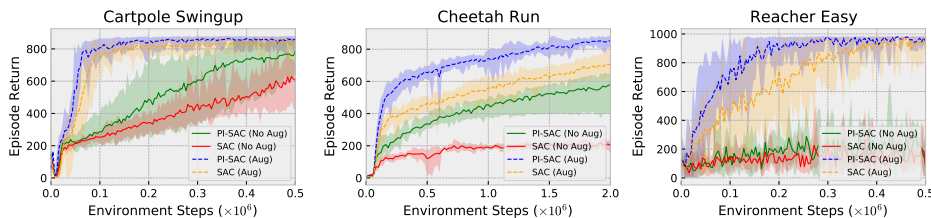

Figure 3: The predictive information improves performance on Cartpole Swingup and Cheetah Run without any data augmentation. With data augmentation, it continues showing strong improvements over the SAC baseline on all three tasks. We perform 5 runs for PI-SAC and SAC without augmentation. More results are presented in Section F.

than or at least comparable to all the baselines. We report our results on Reacher Easy with one gradient update per environment step, on Cheetah Run with four gradient updates, and the rest with two gradient updates. A comparison of PI-SAC agents with different numbers of gradient updates is available in Section B. The comparison in this section is system-to-system as all baseline methods have their own implementation advantages: SLAC uses much larger networks and 8 context frames, making its wall clock training time multiple times slower; DrQ and CURL's SAC differs substantially from the standard SAC [16], including having much larger actor and critic networks; Dreamer is a model-based method that uses RNNs and learns a policy in simulation.

## 4.2 Predictive Information

We test our hypothesis that predictive information is the source of the sample efficiency gains here.

**Data Augmentation.** We follow [27] to train our models with image sequences randomly shifted by $[-4, 4]$ pixels. Without this perturbation, Figure 3 shows that learning the predictive information by itself still greatly improves agents' performance on Cartpole and Cheetah but makes little difference on Reacher. Learning PI-SAC with data augmentation continues showing strong improvements over the SAC baseline with data augmentation and solves all benchmark tasks.

[27, 28] showed that input perturbation facilitates actor-critic learning, and we show that it also improves CEB learning. As described in Section 2, we use the contrastive CatGen formulation to get a variational lower bound on $I(Y; Z)$. Because of its contrastive nature, CatGen can ignore information that is not required for it to distinguish different samples and still saturate its bound. In our experiments without data augmentation, we found that it ignores essential information for solving

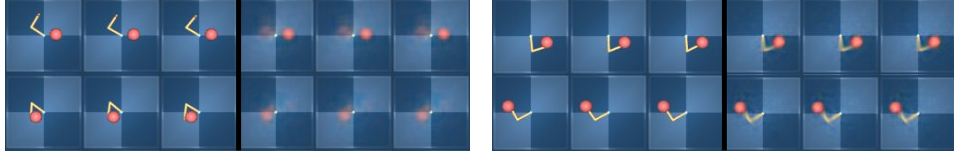

Figure 4: We learn a diagnostic deconvolutional decoder to predict future observations from CEB representations learned along with PI-SAC for Reacher. We show ground truth future observations and the predicted future observations from CEB representations. **Left:** CEB representations learned **without** data augmentation only capture positions of the target. **Right:** CEB representations learned **with** data augmentation capture both the target and the arm.

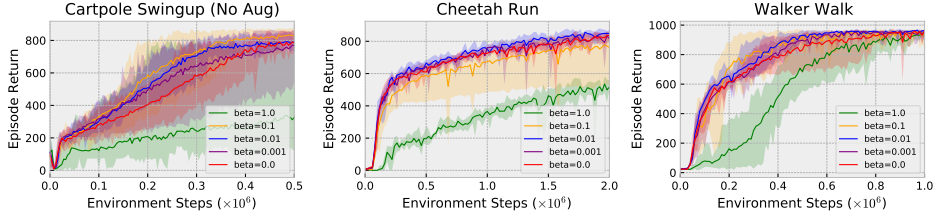

Figure 5: Compression improves agents' performance. We disable data augmentation for Cartpole Swingup to amplify the benefit of compression. At $\beta = 0$, compression is not part of the learning objective. We perform 5 runs for each curve shown in this figure.

Reacher. We train a deconvolutional decoder to diagnostically predict future frames from CEB representations (we stop gradients from the decoder). As shown in Figure 4, CEB representations learned without input perturbation completely fail to capture the arm's pose. This is because CatGen can perfectly distinguish frame sequences in a minibatch sampled from the replay buffer by only looking at the position of the target, since that is constant in each episode but varies between episodes. In contrast, CatGen representations learned with randomly shifted images successfully capture both the target and the arm. This observation suggests that appropriate data augmentation helps CatGen to capture meaningful information for control.

**Compression.** As described in Section 2, we compress the residual information $I(X; Z|Y)$ out to preserve the minimum necessary predictive information. Figure 5 studies the trade-off between strength of compression and agents' performance by sweeping $\beta$ values. Some amount of compression improves sample efficiency and stability of the results, but overly strong compression can be harmful. The impact of $\beta$ on the agent's performance confirms that, even though the CEB representation isn't being used directly by the agent, the auxiliary CEB objective is able to substantially change the agent's weights. Sweeping $\beta$ allows us to explore the frontier of the agent's performance, as well as the Pareto-optimal frontier of the CEB objective as usual [7]. For example, for Cheetah Run, the residual information at the end of training ranges between $\sim 0$ nats for $\beta = 1$, to $\sim 947$ nats for $\beta = 0$. For the top performing agent, $\beta = 0.01$, the residual information was $\sim 6$ nats.

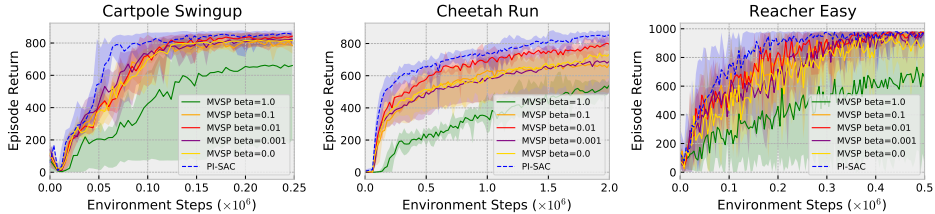

Figure 6: Learning the predictive information outperforms multiview self-prediction (MVSP) at all levels of compression. In this figure we show MVSP agents that predict future rewards conditioning on actions. The MVSP curves show results of 5 runs.

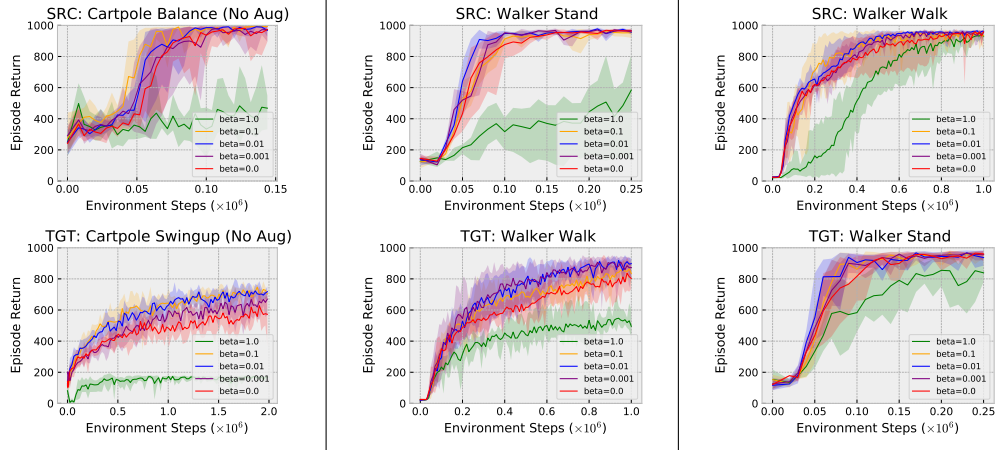

Figure 7: Compression improves task transfer. We train a PI-SAC agent on a source task (SRC), freeze the representation $z_e^{det}$ (see Figure 1), and train a new agent on a target task (TGT). We show the results of normal PI-SAC agents with different $\beta$ for the source tasks, and the transfer results using representations learned on the source tasks with different $\beta$ for the target tasks. Compression substantially improves task transfer when the target task is intuitively more difficult than the source, i.e. Cartpole Balance to Swingup and Walker Stand to Walk. The difference is less significant when the target task is presumably easier, i.e. Walker Walk to Stand. We disable data augmentation for the Cartpole experiment to amplify the benefit of compression. All curves show 5 runs.

**Comparison to Multiview Self-Prediction.** Multiview Self-Prediction (MVSP) is an auxiliary objective used by CURL [39]. CURL uses the InfoNCE bound to capture the mutual information between two random crops $I(X_{crop1}; X_{crop2})$ as an auxiliary task for continuous control from pixels. This approach preserves information about the present, differing philosophically from PI-SAC which captures information about the future. By changing the CEB prediction target from the future ($Y$) to a random shift of the past observation, $X'$, we can achieve the equivalent multiview self-prediction in our framework and fairly compare the two approaches. Figure 6 shows that PI-SAC agents outperform MVSP agents at all levels of compression. But similar to PI-SAC, compression also helps MVSP agents on tasks like Cheetah and Reacher. This empirical evidence suggests that, for RL agents, knowing what will happen in the future matters more than knowing what has happened in the past. Note that Figure 6 shows MVSP agents that predict future rewards conditioning on actions to enable a direct comparison to PI-SAC agents. More results of PI-SAC and MVSP agents with and without future reward prediction on all tasks can be found in Section D.

### 4.3 Generalization to Unseen Tasks

It is well-known that compressed representations can generalize better in many machine learning settings [36, 3, 8], including RL [20, 12, 11]. In addition to sample efficiency, for more testing of generalization, we explore transferring representations to an unseen task with the same environment dynamics. Specifically, we learn a PI-SAC agent on a source task, freeze the representation $z_e^{det}$ (see Figure 1), and train a new agent for a target task using the frozen representation. Figure 7 shows that compressed representations generalize substantially better to unseen tasks than uncompressed representations. Especially when the target task is intuitively harder than the source task, i.e. Cartpole Balance to Swingup and Walker Stand to Walk, the performance differences between different levels of compression are more significant on the target task than on the original tasks. It is, however, less prominent when the target task is easier, i.e. Walker Walk to Stand. Our conjecture is that solving the original Walk task would require exploring a wider range of the environment dynamics that presumably includes much what the Stand task would need. On the other hand, transferring from Stand to Walk requires generalization to more unseen part of the environment dynamics. Note that in these settings it is still more sample efficient to train a full new PI-SAC agent on the target task. These experiments simply demonstrate that the more compressed predictive information models have representations that are more useful in a task transfer setting.

# 5   Related Work

**Future Prediction in RL.**   Future prediction is commonly used in RL in a few different ways. Model-based RL algorithms build world model(s) to predict the future conditioned on the past and actions, and then act through planning [6, 14, 17, 18, 23, 25, 41]. Some of the curiosity learning approaches reward an agent based on future prediction error or uncertainty [5, 31, 40]. This work falls in another group, where future prediction is used as an auxiliary or representation learning method for model-free RL agents [1, 10, 14, 22, 29, 30, 33, 34, 35, 37]. We hypothesize that the success of these methods comes from the predictive information they capture. In contrast to prior work, our approach directly measures and compresses the predictive information, so that the representation avoids capturing the large amount of information in the past that is irrelevant to the future. As described in Section 2, the predictive information that we consider captures environment dynamics. This is different from some other approaches [1, 30] that use contrastive methods (e.g. InfoNCE) to capture temporal coherence of observations instead of dynamics (since their predictions are not action-conditioned) and thus have their limitations in off-policy learning. See Section E for discussion and results without conditioning on actions.

**Other Types of Representation for RL.**   Besides future prediction, previous contributions have investigated other representation learning methods for RL. An example is inverse dynamics [5, 31] which learns representations through predicting actions given the current and next observed states. In contrast to future prediction, inverse dynamics only reflects what the agent can immediately affect, and thus may not be sufficient [5]. Another example is learning representations through retaining information about the past observations using reconstructive or contrastive approaches [39, 43], but that does not lean toward capturing information relevant for achieving the future and goal. We show that future prediction is empirically superior in Section 4.2 and Section D.

**Continuous Control from Pixels.**   Recent successes in continuous control from visual observations can roughly be classified into two groups: model-based and model-free approaches. Prominent model-based approaches like PlaNet [18] and Dreamer [17], for example, learn latent dynamics and perform planning in latent space. Several recent works with the model-free approach [27, 39, 28] have demonstrated that image augmentation improves sample-efficiency and robustness of model-free continuous control from pixels. In this work, we also perform image augmentation and discover its value in learning the predictive information with a contrastive loss (Section 4.2). An example falling between model-based and model-free is SLAC [29] which captures latent dynamics for representation learning but learns the model-free SAC [16] with the representation. The approach is different from ours as PI-SAC does not model roll-outs in latent space.

# 6   Conclusion

We presented Predictive Information Soft Actor-Critic (PI-SAC), a continuous control algorithm that trains a SAC agent using an auxiliary objective that learns a compressed representation of the predictive information of the RL environment dynamics. We showed with extensive experiments that learning a compressed predictive information representation can substantially improve sample efficiency and training stability at no cost to final agent performance. Furthermore, we gave preliminary indications that compressed representations can generalize better than uncompressed representations at task transfer. Future work will explore variations of the PI-SAC architecture, such as using RNNs for environments that require long-term planning.

## Broader Impact

This work attempts to expand the applicability of RL to visual inputs and expand the range of RL applications, especially in Robotics. In its current form this work is applied to simulated environments, but thanks to the improvements in sample efficiency and generalization it increases the possibility of training directly on real-world robots.

However, advancements in robotic automation likely have complex societal impacts. One potential risk is creating shifts in skill demand and thus structural unemployment. Improving RL autonomous agents' applicability to visual inputs is a potential threat to a range of employment types, for example,

in the manufacturing industry. Public policy and regulation support will be necessary to reduce societal and economic friction as techniques like these are deployed in the physical world. On the other hand, potential positive outcomes from continued RL improvements include replacement of human workers at high-risk workspace, reduction of repetitive operations, and productivity increases.

We see opportunities that researchers and engineers would benefit from adding CEB and the Predictive Information to the list of tools in RL. We would like to note that designing RL tasks and reward functions can have potential biases if applied to real systems that interact with users. Therefore we encourage further research to understand the impacts, implications, and limitations of using this work in real-world scenarios.

## Acknowledgments and Disclosure of Funding

We thank Justin Fu, Anoop Korattikara, and Ed Chi for valuable discussions. Our thanks also go to Toby Boyd for his help in making the code open source. Furthermore, we thank Danijar Hafner, Alex Lee, Ilya Kostrikov and Denis Yarats for sharing performance data for the Dreamer [17], SLAC [29], and DrQ [27] baselines.

John Canny is associated with both Google Research and University of California, Berkeley. Honglak Lee is associated with both Google Research and University of Michigan. Honglak would like to acknowledge NSF CAREER IIS-1453651 for partial support.

## Footnotes

[1] https://github.com/google-research/pisac

[3]SLAC, Dreamer, and DrQ learning curves were provided to us by the authors.

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
