[Supplementary Material]

# Appendices

## A   PI-SAC Implementation

**Initial CEB training steps.**   After collecting initial experiences with a random policy (in Algorithm 1), we optionally pre-train with the predictive information CEB objective ($\theta_e, \bar{\theta}_e, \psi_e, \psi_b$ are updated). The amount of initial CEB steps are selected empirically for each task (listed in Table 2).

**Observation Horizon and Frame Stacking.**   As described in Section 3, we limit our observations from $-T + 1$ to $T$ (following [39, 27], we set $T = 3$). We construct the observational input to the encoder and backward encoder as a $T$-stack of consecutive frames, where each frame is a $[84 \times 84 \times 3]$ RGB image rendered from the 0th DMControl camera. The pixel values range from $[0, 255]$, and we divide each pixel by 255.0.

**Evaluation Setups.**   We evaluate our agent at every evaluation point by computing the average episode return over 10 evaluation episodes. At test time, our policy is deterministic and uses the mean of the policy distribution. For most of the experiments, we evaluate every 2500 environment steps after applying action repeat for Cheetah, Walker, and Hopper tasks. For Ball in Cup, Cartpole, Finger, and Reacher tasks, we evaluate every 1000 environment steps after applying action repeat.

**SAC Implementation.**   Our SAC implementation is based off of TF-Agents [13]. It follows the standard SAC implementation [16]. The performance and sample-efficiency match with the benchmark results reported in [16].

### A.1   Network Architecture

**Encoder Networks.**   The convolutional encoder architecture consists of four convolution layers with $3 \times 3$ kernels, 32 channels, similar to the encoder architecture being used in [43, 39, 27]. We use stride 2 at the first convolution layer and 1 in the rest. Filter Response Normalization and Thresholded Linear Unit [38] are applied after each convolution layer. The output of the last convolution layer is fed into a fully-connected layer which projects to a 50-d feature vector and followed by Layer Normalization [2]. This gives us the 50-d $z_e^{\text{det}}$. As shown in Figure 1, we stop gradients from the actor network, but allow the critic optimizer and the CEB optimizer to update the convolutional encoder.

**Actor and Critic Networks.**   Implementations of actor and critic follow the standard SAC [16]. Both actor and critic are parameterized by MLPs with two 256-d hidden layers. The actor network outputs mean and covariance for a parametric Gaussian distribution. And we use $\texttt{tanh}$ as an invertible squashing function to enforce the action bounds as in [16]. Inputs to the critic network is a concatenation of $z_e^{\text{det}}$ and action.

**CEB Auxiliary Model.**   Our CEB forward and backward MLP encoders are parameterized by MLPs with two 128-d hidden layers. Each MLP outputs 50-d mean followed by Batch Normalization [21] for a multivariate Gaussian distribution, and we fix diagonal covariance at $1.0$. Inputs to the forward MLP is a concatenation of $z_e^{\text{det}}$ and $T$ future actions. Inputs to the backward MLP is a concatenation of $z_e^{\text{det}}$ and $T$ future rewards.

### A.2   Hyperparameters

Throughout these experiments we mostly use the standard SAC hyperparameters [16]. The hyperparameters that are fixed across all tasks are listed in Table 2. The size of the replay buffer is a smaller $10^5$ due to high memory usage for storing image observations. The heuristic entropy target is set to $-dim(A)/2$, a default value used in TF-Agents SAC implementation [13], where $dim(A)$ is number of dimensions of action. In our experiments the results are similar to using $-dim(A)$, the default used in [16].

The amount of action repeat (described in Section 4), initial collection steps, and initial CEB steps for each task are listed in Table 2. The standard SAC takes 10,000 initial collection steps, but for some of the tasks we take fewer steps of 1,000 in favor of sample efficiency.

Table 2: **Left:** Global PI-SAC hyperparameters. **Right:** Per-task PI-SAC hyperparameters. PlaNet tasks are indicated with (P).

| Parameter | Value |
|---|---|
| optimizer | Adam [26] |
| batch size | 256 |
| actor learning rate $\lambda_\pi$ | $3 \times 10^{-4}$ |
| critic learning rate $\lambda_Q$ | $3 \times 10^{-4}$ |
| alpha learning rate $\lambda_\alpha$ | $3 \times 10^{-4}$ |
| CEB learning rate $\lambda_{CEB}$ | $3 \times 10^{-4}$ |
| discount ($\gamma$) | 0.99 |
| replay buffer size | $10^5$ |
| entropy target $\mathcal{H}$ | $-dim(A)/2$ |
| target smoothing coefficient ($\tau$) | 0.005 |
| target update interval | 1 |
| initial $log(\alpha)$ | 0.0 |
| backward encoder EMA update rate | 0.05 |
| observation horizon T | 3 |

| Task | Action Repeat | Initial Collection Steps | Initial CEB steps |
|---|---|---|---|
| Cartpole Swingup (P) | 4 | 1000 | 5000 |
| Cartpole Balance Sparse | 2 | 1000 | 5000 |
| Reacher Easy (P) | 4 | 1000 | 5000 |
| Ball in Cup Catch (P) | 4 | 1000 | 5000 |
| Finger Spin (P) | 1 | 10000 | 0 |
| Cheetah Run (P) | 4 | 10000 | 10000 |
| Walker Walk (P) | 2 | 10000 | 10000 |
| Walker Stand | 2 | 10000 | 10000 |
| Hopper Stand | 2 | 10000 | 10000 |

# B    PI-SAC and SAC at Different Numbers of Gradient Steps

Figure 8: Comparison of PI-SAC and SAC with image augmentation at different numbers of gradient steps (gs) per environment step. We report results at 1 and 2 gradient steps, except that we show 1 and 4 gradient steps for Cheetah Run. PI-SAC consistently outperforms the SAC baseline.

Another way to improve sample efficiency in SAC and PI-SAC models is to increase the number of gradient steps taken per environment step collected. In Figure 8, we see that PI-SAC outperforms the SAC baseline while varying gradient steps, particularly on Ball In Cup Catch and Cheetah Run.

# C   Comparison to SAC from States

Figure 9: Comparison of PI-SAC (from pixels) to SAC from states (state-SAC) at different numbers of gradient steps (gs) per environment step. We report results at 1 and 2 gradient steps for all tasks except Cheetah Run, which uses 1 and 4 gradient steps. We use the action repeat values from Table 2 for state-SAC as well. PI-SAC performs comparably to state-SAC on most tasks.

It is interesting to compare PI-SAC agents, which are trained from pixels, to SAC agents that have been trained from states. Generally, training from pixels is considered to be more challenging than training from states. However, we find that PI-SAC performs comparably to SAC from states on most tasks. On Cheetah Run and Finger Spin, PI-SAC significantly outperforms state-SAC, indicating that those tasks benefit strongly from representations that model what will happen next, rather than simply needing a precise description of the current state. In contrast, state-SAC has a noticeable sample efficiency advantage over PI-SAC on Ball In Cup Catch, indicating that a precise description of the current state that is stable throughout training is more important than learning to model what will happen next.

# D  Comparison of PI-SAC and Multiview Self-Prediction Agents

Figure 10: Learning the predictive information outperforms multiview self-prediction (MVSP), which is described in Section 4.2. We compare PI-SAC to using the MVSP auxiliary task on all 9 tasks. Specifically, we show PI-SAC and MVSP results with and without predicting future rewards in the auxiliary task. We use the default $\beta = 0.01$ for experiments in this figure.

In Figure 10 we present complete results of PI-SAC and Multiview Self-Prediction (MVSP) agents with and without predicting future rewards in the auxiliary task as an extension to Figure 6. For these experiments, we don't sweep $\beta$ for the MVSP models, and instead use the default $\beta = 0.01$ for all models. In all cases, the PI-SAC models achieve equal or better performance. In general predicting future rewards leads to performance improvements.

# E  The Importance of Conditioning on Actions

Figure 11: Standard PI-SAC that models environment dynamics $p(s'|s, a)$ outperforms the version that models $p(s'|s)$ without conditioning on actions on most tasks. We use the default $\beta = 0.01$ for experiments in this figure.

In Figure 11, we compare PI-SAC agents that predict the future conditioning on actions and those are not conditioning on actions to shed lights on the importance of modeling environment dynamics instead of temporal coherence. The former ones model environment dynamics $p(s'|s, a)$ which is independent of policy, whereas the latter ones instead capture temporal coherence of observations, i.e. the marginal $p(s'|s)$. Obviously, modeling $p(s'|s)$ has its limitation in off-policy learning as it fits to experience replay instead of the current policy [30]. As shown in Figure 11, conditioning on actions empirically leads to better sample efficiency and/or returns on most tasks, except for being slightly slower on Cartpole Balance Sparse. For the Cartpole Balance Sparse task, our conjecture is that, because the $p(s'|s)$ model is a simpler one without needing to consider actions and the Cartpole starts right at where the goal is in this task, it is still able to learn information as useful for achieving the goal as the $p(s'|s, a)$ model and even slightly faster.

# F   PI-SAC and SAC without Image Augmentation

Figure 12: Comparison of PI-SAC and SAC both without image augmentation on all 9 tasks. We perform 5 runs for experiments without image augmentation. PI-SAC without image augmentation always matches or improves on the SAC baseline, but some tasks are only solved with the addition of image augmentation.

In Figure 12 we compare PI-SAC agents with SAC and PI-SAC agents trained without image augmentation on all nine tasks. This is the same setting as Figure 3. Learning the predictive information without image augmentation is sufficient to significantly improve SAC agents for some tasks, and is never detrimental compared to the SAC baseline. However, augmentation is essential to solving Reacher Easy, Walker Walk, and Hopper Stand. On all tasks, having both the predictive information and image augmentation performs the best.

# G  Comparison of Contrastive and Generative PI-SAC

Figure 13: Comparison of the standard contrastive version and PI-SAC and the generative version of PI-SAC which directly predicts future observations and future rewards. The contrastive version shows better sample-efficiency and performance than the generative version every task except Hopper Stand, where the two approaches are essentially indistinguishable. The wall time per gradient step of the contrastive models is about 30% faster than the generative models, even with the small number of frames being predicted (3 future frames).

As shown in [7], under an assumption of a uniform distribution over the training examples, the contrastive CatGen formulation (eq. (5)) approximates the decoder distribution:

$$\frac{b(z|y)}{\sum_{k=1}^{K} b(z|y^k)} \approx p(y|z) \tag{11}$$

Instead of using CatGen in CEB, we can alternatively learn to predict $y$, the future observations and future rewards. This gives a generative variant of PI-SAC. To predict the future observations for the generative PI-SAC, we use a decoder network consisting of four transposed convolution layers with features of $(12, 64, 32, 3)$, kernel widths of $(3 \times 3, 3 \times 3, 11 \times 11, 3 \times 3)$, and strides of $(2, 2, 1, 2)$. To predict the future rewards, we use an MLP with 50-d first hidden layer and 25-d second hidden layer.

Figure 13 compares the standard contrastive version of PI-SAC to the generative variant. It shows that the contrastive version is generally more sample-efficient and gives better performance. Additionally, the generative version is slower to train in terms of wall time. These observations lead us to prefer the contrastive CatGen formulation for PI-SAC.

Figure 14: **Left:** Actor Representation PI-SAC system diagram. **Right:** Critic Representation PI-SAC system diagram.

# H   Representation PI-SAC

The CEB representation, $Z$, can be used directly by either the actor, the critic, or both. Figure 14 show the system diagrams for actor and critic Representation PI-SAC models. For models that use CEB representations for both actor and critic, it suffices to combine those systems so that there are two separate CEB objectives, but with both objectives updating the same convolutional encoder parameters ($\theta_e$). In this setting, neither the actor nor the critic pass gradients back through the convolutional encoder parameters. This an important difference from PI-SAC, where both the CEB objective and the SAC critic objective contribute gradient information to the convolutional encoder parameters. As shown in Figure 14, the actor or the critic use the mean of the forward encoder's $Z$ distribution ($z_e^{\text{det}}$ in the figures). It is also possible to train using samples from the distribution ($z_0$ in the figures). Empirically, we found that the results were qualitatively the same, but often with higher variance in evaluation performance early in training, so we only present results using the mean of the representation here.

Figure 15 shows results comparing Representation PI-SAC models to Dreamer on the same tasks from the paper. In these experiments, no hyperparameters are changed between the different tasks. All Representation PI-SAC models in Figure 15 are trained using the hyperparameters in Table 2, action repeat of 2, 1000 initial collection steps, and 0 initial CEB steps. This makes the results directly comparable to Dreamer, which also uses action repeat of 2 for all tasks and has no task-specific hyperparameters.

Using a CEB representation for the actor, the critic, or both still gives much better sample efficiency than Dreamer at 6 of the 9 tasks, but the lack of the full set of future actions as input to the forward encoder in these models appears to make it more difficult for the agents to solve locomotion tasks like Cheetah Run and Walker Walk. We also tested a critic Representation PI-SAC model variant that allowed critic gradients to flow to the convolutional encoder parameters (as in PI-SAC), and found that doing so improved performance on Cartpole Swingup and Reacher Easy, but substantially worse on Ball In Cup Catch, Cartpole Balance Sparse, and Cheetah Run, and comparable on the remaining four tasks. This indicates that the difficulty on the locomotion tasks we see here is not due to the lack of critic gradients.

On most tasks, we found that actor Representation PI-SAC had the strongest performance and best stability of the three Representation PI-SAC variants. In particular, its stability on Cartpole Balance Sparse was remarkable: from 80,000 environment steps through 480,000 environment steps, all five agents got perfect scores of 1,000 on all 10 evaluation trials that occurred every 10,000 environment steps, for a total of 2,000 perfect evaluations in a row. In contrast, the other two variants (and Dreamer) had substantial deviations from perfect scores throughout training. More exploration is necessary to understand why actor Representation PI-SAC has this stability advantage.

Figure 15: Representation PI-SAC models with action repeat of 2 and no task-dependent hyperparameter changes, making the experiments directly comparable to the Dreamer results. All curves are 5 runs.

# I Discussion of Compression

As discussed in Section 4.3, compression is known to improve generalization [36, 3, 8]. For example, in [36], the authors show that every additional bit in a representation $Z$ requires four times as much training data to achieve the same generalization, which may explain PI-SAC sample efficiency gains:

$$|I(Y;Z) - I(\hat{Y};Z)| \leq \mathcal{O}\left(\frac{|Y|2^{I(\hat{X};Z)}}{\sqrt{N}}\right) \qquad (12)$$

Here, $\hat{X}$ and $\hat{Y}$ are the training observations and targets, $Z$ is the learned representation, $N$ is the number of training examples, and $X$ and $Y$ are the observations and targets in the full distribution the training examples are sampled from. Since we are trying to learn $Z$ that maximizes $I(Y;Z)$ while only observing $\hat{X}$ and $\hat{Y}$, we want $I(\hat{Y};Z)$ to be as close to $I(Y;Z)$ as possible. This bound makes it clear that we can make the two close by either increasing $N$ or decreasing $I(\hat{X};Z)$.

Of course, one way to make that bound tight is to make $Z$ independent of $\hat{X}$ and $\hat{Y}$. In that case, all of $I(\hat{X};Z)$, $I(\hat{Y};Z)$, and $I(Y;Z)$ converge to 0. To avoid this, we would like to learn the *Minimum Necessary Information* (MNI), introduced in [7]. The MNI is defined as the equality:

$$I(X;Y) = I(X;Z) = I(Y;Z) \qquad (13)$$

Intuitively, this says that we are trying to find a representation $Z$ that captures exactly the information that is shared between $X$ and $Y$. Learning such a $Z$ corresponds to finding the maximum of $I(Y;Z)$ while also having $I(X;Z|Y) = 0$.[4] This is basis of the CEB objective function which we apply here to the problem of bounding the Predictive Information, as described in Section 2.

## Footnotes

[4] And also having $I(Y;Z|X) = 0$, but that can be enforced by having the Markov chain $Z \leftarrow X \rightarrow Y$, which means that $Z$ is a stochastic function of $X$ only.