[Reviews · NeurIPS 2020]

Review 1

Summary and Contributions: This paper aims to model the predictive information, namely the mutual information between the past and the future. The proposed method, PI-SAC, learns a policy and a compressed, predictive representation that makes the environment more Markovian by an auxiliary loss, simultaneously.

Strengths: - PI-SAC achieves superior or comparable performance in terms of sample complexity and episode return compared to many competitive baselines. - Future prediction has been an important ingredient in many prior works before, but unlike them, PI-SAC directly models and compresses the predictive information.

Weaknesses: - While the predictive information does improve performance over those without this auxiliary loss, it appears that a major portion of the performance gain still comes from data augmentation (Figure 3). SLAC, which does not use data augmentation, clearly outperforms PI-SAC (No Aug) in some tasks.

Correctness: - Line 124-125: "PI-SAC consistently achieves state-of-the-art performance and better sample ef´Čüciency across all benchmark tasks" might be an overclaim. Judging from Figure 2, I believe PI-SAC does not really have better sample complexity than DrQ in Walker Walk, Cartpole Balance Sparse and Hopper Stand. - Figure 7: Perhaps I'm missing something, but I can't see how these results suggest the more compressed the representation is, the more generalizable it is. There doesn't seem to be any significant difference of performance drop for both beta=0.1 and beta=0.0 when transferred to the target task.

Clarity: Overall, the paper is relatively well-written, but can definitely still be improved for clarity. It might be useful to spend more time to explain CEB (e.g., what the variational backwards encoder is?). Additionally, when explaining PI-SAC in Section 3, it is a bit difficult to follow. My suggestion is to perhaps make use of Figure 1 to provide a high-level overview of how the algorithm works.

Relation to Prior Work: This paper clearly states how this approach differs from previous works. Unlike CURL that captures the mutual information between two random crops using the InfoNCE bound, PI-SAC aims to capture the predictive information between the past and the future using CEB. Although there has been prior works that use future prediction as an auxiliary objective, they did not directly measure or compress the predictive information.

Reproducibility: Yes

Additional Feedback: - Can you explain the intuition behind the choice of X and Y in your setting? - In Section 4.3, generalization to unseen dynamics is discussed. I'm wondering if you have tried this on different environments, or to varying degrees of different dynamics? Typos: - Line 17: "there is common a Markov assumption" --> "there is a common Markov assumption"


Review 2

Summary and Contributions: In this work the authors add an additional surrogate task into the SAC algorithm. The surrogate task is to minimize the future prediction loss, which is learned via the conditional entropy bottleneck objective. ** I read the author response and decided to keep my score. Thanks!

Strengths: The paper is generally well written is easy to follow. Furthermore the idea is simple and coherent in my opinion. The experiments try to empirically investigate different aspects of the technique and shown to give good performance relatively to existing methods. To the best of my knowledge, although there has been several works that used predictive information in RL (for example, in curiosity driven RL) the approach investigated by the authors is novel.

Weaknesses: -) Although the technique is interesting, it does result in a final performance which is comparable to existing approaches (although better by a bit). I would love to see cases in which existing approaches fail, however, using this technique allows to get a good performance. -) Beside of testing the algorithm, the paper does not motivate this method from a theoretical perspective nor relates it to existing algorithms in theoretical RL. For this reason, it is hard not to wonder if this approach works because it is generally good, or only due to the specific types of problems on which it was tested on.

Correctness: To the best of my knowledge, the empirical methodology is correct. Especially since the authors (as they claim) did not change hyperparameters of existing algorithms. For this reason, it seems their approach is robust.

Clarity: The paper is well written.

Relation to Prior Work: I'm not an expert in this field, however, I am aware to several missing references, e.g., https://link.springer.com/content/pdf/10.1007/s12064-011-0142-z.pdf https://arxiv.org/abs/1705.05363 https://arxiv.org/pdf/1808.04355.pdf The authors should highlight the difference between their approach and curiosity driven approach as the two are very closely related.

Reproducibility: Yes

Additional Feedback: * Comment on the predictive information. In systems we cannot control, I(past; future) only depends on the distribution on the random variables. However, How do you define it when an agent can control some portions of the systems? In this situation, the future depends on the policy of the agent. I think, for this reason, a bit more care should be taken when using I(past; future) is RL. *Experiments. Although the experiments suggest PI-SAC is better, it does not perform dramatically better than existing methods (Figure 2).


Review 3

Summary and Contributions: This paper proposes a new type of auxiliary task to learn better representations in RL tasks. The idea is to use past observations and future actions to predict future states and rewards. The paper also experiments with a regularization that minimizes mutual information between input and representation.

Strengths: The proposed auxiliary task is well motivated. It makes intuitive sense to learn a representation that encodes the minimal information to predict the future states The experiments cover a lot of different situations, including with/without data augmentation, the best amount of mutual information minimization, generalization of the learned representation.

Weaknesses: The idea of using an auxiliary prediction task has been explored by several papers in the literature (which the authors cited). Unfortunately this means a large number of potential baselines. For example, future reward prediction has already been used as an auxiliary task in [1]. My main concern is that the experiments mainly demonstrate the benefit of using an auxiliary task, while the main claimed contribution is a new type of auxiliary task. The paper compares with one such baseline (MVSP) which was used in a different setup (CURL with unsupervised representation learning). Even though experiments cannot be all inclusive, at least one would expect comparison with one compelling baseline on the same problem as the original paper. All the experiments are on 6 out of the ~15 tasks in the Deepmind control suite. To instill confidence that the improvement is statistically significant, it is better to use the entire dataset, or explain how this subset is chosen.

Correctness: I did not find major issues. I had some questions about the empirical methodology in the weakness section.

Clarity: The paper is quite clear.

Relation to Prior Work: I find the discussion sufficient.

Reproducibility: Yes

Additional Feedback: After reading the rebuttal, I think the comparison and empirical demonstration of the superiority of the proposed approach is somewhat sufficient (at least on some popular benchmarks). Overall I think this is a small but empirically well justified improvement and I didn't notice any major flaws.


Review 4

Summary and Contributions: The paper makes use of the Conditional Entropy Bottleneck (CEB) objective as published recently in a single task RL setting; specifically, for problems relating to control from pixel data. The paper enhances the vanilla SAC objective with the CEB as an auxiliary task. It compares with other SOTA model-free and model-based RL methods. In particular, the representation gained through the CEB objective has a notion of the predictive future, which, through experiments, has been shown to be useful. The ablations show, as far as I can tell, that the compression and the particular way they do future prediction are useful.

Strengths: - The paper makes use of a novel idea of CEB, which was published recently, and applies it to single task RL. - The experiments were well crafted, and the ablation tests were useful. Furthermore, the experiments were carried out on a wide variety of environments, with different complexities. - The supplementary material contains further experiments that were interesting to read. - The language in the paper is well written, and pleasant to read. - The broader impact section was well written.

Weaknesses: - It was difficult to understand the model that they used; in my opinion, this was because I think Section 3 could have been written more clearly. See clarity section for more notes.

Correctness: - They make use of existing techniques, so based on their validity, these appear correct. For example, Equations 1-5 can be found from their reference [6]. - The experiments appear to have been carried out well. They compare with 3 other SOTA models, which were chosen to represent different model-free and model-based methods. In particular, 10 repeated experiments were run for each case. - From these experiments, it does appear that there is a clear benefit to this model.

Clarity: - Why were 2 critic networks used? I don't believe an explanation for this was clearly written. Further, 2 critics don't appear in Figure 1, which was confusing. (maybe I missed something?) - Perhaps the inputs and outputs of the different functions can be written more clearly. For example, in Figure 1, the input to the actor pi appears to be z, the representation from the convolutional encoder. But the notation states that is \pi is conditioned on the pixel states. Furthermore, \pi in most places takes as input the state, but in Figure 1, it takes in s_{-2:0}. Aren't these of different inputs, since the latter is 3x the 'dimension' of the other. - This is a bit nit-picky, but could \phi be added to \pi on the right hand sides of Equation 8, and Line 76; this is to show where the dependence to \phi is. - I'm not sure the Data Augmentation subsection in Section 4.2 was particularly useful. In my understanding, it seems to show that Data augmentation is useful in both PI-SAC and SAC, but doesn't really give much more information regarding the usefulness of PI-SAC. (for example, SAC-aug appears to perform better than PI-SAC-no-aug.) - I wonder if the Generalisation section could be left in the supplementary material, since they are using a version of the model (Representation PI-SAC) that isn't used in the rest of the experiments. Perhaps the extra space from this can be used to explain their model in more detail.

Relation to Prior Work: The paper makes use of the CEB, and SAC, which the authors have discussed well. There is also a discussion of how future prediction is used in RL, and other works that deal with pixels to control.

Reproducibility: Yes

Additional Feedback:

[Author Response · NeurIPS 2020]

We sincerely thank our reviewers for the valuable feedback. We note the consensus around the technical novelty of
learning compressed representations of the predictive information, the strong empirical performance with comprehensive
evaluations, and the clear rationale and presentation. For reproducibility, we plan to release our code by Oct. 1.

**[R1, R2] Improvements over previous methods and the SotA claim:** Regarding the SotA claim, we will clarify in
revision that PI-SAC is better than or at least comparable to any previous SotA for all tasks we evaluated. Additionally,
we think the perception that PI-SAC is only slightly better than previous methods is partially a presentation issue. To
clarify the differences in performance, the table below is the same PlaNet benchmark comparison table used in both
DrQ and CURL. It clearly shows the substantial benefit of PI-SAC. The full table will be included in the revision to
augment Fig. 2.

**[R3] Comparison to auxiliary baselines:** We did not include
CURL in our submission due to a critical reporting error in the
CURL v1 paper (compare the v1 and v3 versions on arxiv). Now
that the CURL results have been corrected, we will include them.
The table to the right shows that PI-SAC clearly outperforms
CURL. Besides MVSP, we also compare to uncompressed PI-SAC
since all of the other auxiliary future prediction tasks that we are
aware of in the literature do not attempt to explicitly compress
the predictive information. In appendix F, we compare to explicit
future prediction using generative models and explain that those

| 100k step scores | PI-SAC | CURL | DrQ |
|---|---|---|---|
| Ball in Cup Catch | **933±16** | $769 \pm 43$ | $913 \pm 53$ |
| Cartpole Swingup | **816±72** | $582 \pm 146$ | $759 \pm 92$ |
| Finger Spin | **957±45** | $767 \pm 56$ | $901 \pm 104$ |
| Reacher Easy | **758±167** | $538 \pm 233$ | $601 \pm 213$ |
| Walker Stand | **942±21** | N/A | $832 \pm 259$ |
| 500k step scores | PI-SAC | CURL | DrQ |
| Cheetah Run | **801±23** | $518 \pm 28$ | $660 \pm 96$ |
| Hopper Stand | **821±166** | N/A | $750 \pm 140$ |
| Walker Walk | **934±53** | $902 \pm 43$ | $921 \pm 45$ |

are also maximizing MI. Finally, as mentioned in Sec. 3, we include future rewards as part of $Y$. We have updated the
paper with an ablation removing reward prediction. It slightly degrades PI-SAC performance.

**[R1, R4] Comparing PI-SAC(No Aug) to SAC(Aug) and SLAC:** Sec. 4.2 (from line 142) and Fig. 4 explain why
CatGen fails without augmentation. PI-SAC(No Aug) is showing a failure mode; it is not meant to be compared with
SAC(Aug) or SLAC. PI-SAC's benefit is demonstrated with the substantial difference between PI-SAC(Aug) and
SAC(Aug) in Fig. 3. Also note that SLAC is a completely different system that uses much larger networks and 8 context
frames. SLAC's wall clock training time is ∼5x slower than PI-SAC. Comparison to SLAC and the other baselines
can only be done at a full systems level due to these major differences. It's plausible that SLAC (and Dreamer) would
benefit from data augmentation, but PI-SAC would also likely benefit from larger networks and more context frames.

**[R1] Generalization:** In Fig. 7 we mistakenly used different axis
scales between figures which obscures the performance difference
between source and target tasks. We fixed the axis scales and up-
dated the experiments to use PI-SAC instead of Representation
PI-SAC for consistency with the other experiments in the main pa-
per. Results for Walker Stand to Walker Walk can be seen to the
right. For dynamics transfer, we varied the testing pole length from
0.4 to 1.6 (trained on 1.0). We find that some compression is always
better than none. We will describe these results in the appendix.

**[R3] Choice of DM Control tasks:** The first 6 tasks (out of 9) are
the PlaNet benchmark (mentioned in line 108). All of the baselines we compare with use this set. We expanded this
set with Walker Stand (for task transfer), Cartpole Balance Sparse (for sparse rewards), and Hopper Stand from the
Dreamer benchmark to further explore PI-SAC's generality.

**[R2] Theoretical motivation and generality:** We explore future prediction from an information-theoretic perspective,
using CEB [6] to measure and compress the predictive information [4]. As we discuss in Sec. 2, PI-SAC is motivated
by the observation that correctly modeling the predictive information requires learning a compressed representation of
the past. Due to space limitations, we refer the reader to those works for detailed theoretical background. In Sec. 5 (line
193), we list previous successes of future prediction for representation learning and auxiliary tasks on various types of
RL problems, which is evidence that PI-SAC should apply more broadly.

**[R1, R2] Representation dependence on policy and choice of X and Y:** The CEB model captures only the environ-
ment dynamics $s, a \rightarrow s'$ (which is independent of the policy) by conditioning the encoder $e(z|x)$ on the future actions
(actions are part of $X$). Part of this is explained around lines 201-204, but we will add clarifications. Following CURL
and DrQ, we use 3 frames for $X$. We make $Y$ symmetric to $X$; it contains the next 3 frames and their rewards.

**[R1, R2, R4] Citations and other clarity questions:** Thanks for suggesting the curiosity papers – we will include
them in Sec. 5. We will make the CEB and PI-SAC descriptions more self-contained, and improve Sec. 3. We will add
descriptions for double critics (they are part of SAC [11]) and improve the notation. We updated the generalization
section to use PI-SAC rather than Representation PI-SAC (see the Walker task transfer figure above).

[Meta-Review · NeurIPS 2020]

This paper experimentally checks the hypothesis that capturing the predictive information is useful in RL. The novelty of the proposed auxiliary task lies in the fact that it learns a *compressed* representation of the predictive information. The experiments are convincing in showing the improvement of PI-SAC over SAC (with or without data augmentation) and over other approaches using auxiliary tasks.